# Effects of Intracellular Polysaccharides and Proteins of *Auxenochlorella pyrenoidosa* on Water Quality, Floc Formation, and Microbial Composition in a Biofloc System

**DOI:** 10.3390/microorganisms13071704

**Published:** 2025-07-21

**Authors:** Mengsha Lou, Yuhan Zhang, Manman Zhang, Hangxian Zhou, Yixiang Zhang, Qiang Sheng, Jianhua Zhao, Qiyou Xu, Rongfei Zhang

**Affiliations:** 1School of Life Science, Huzhou University, Huzhou 313002, China; loumengsha503@163.com (M.L.); 18963264817@163.com (Y.Z.); 19816909937@163.com (M.Z.); zhxjournal@163.com (H.Z.); yxzhang@zjhu.edu.cn (Y.Z.); qsheng@zjhu.edu.cn (Q.S.); zhaojianhua@zjhu.edu.cn (J.Z.); xuqiyou@sina.com (Q.X.); 2Zhejiang Provincial Key Laboratory of Aquatic Resources Conservation and Development, Huzhou University, Huzhou 313002, China; 3National and Local Joint Engineering Laboratory of Aquatic Animal Breeding and Nutrition, Huzhou University, Huzhou 313002, China

**Keywords:** microalgae, intracellular substance, polysaccharides, proteins, microbial composition, flocculation, nitrogen transformation

## Abstract

The use of *Auxenochlorella pyrenoidosa* (formerly *Chlorella pyrenoidosa*) and its intracellular substances (ISs) to promote biofloc development has been extensively studied. To identify the key components of the ISs of *A. pyrenoidosa* that drive biofloc formation, algal-extracted polysaccharides (AEPSs) and algal-extracted proteins (AEPTs) were isolated from the ISs. In this study, we established four groups: ISs, AEPSs, AEPTs, and tap water (TW, control), to investigate the effects of AEPSs and AEPTs on biofloc formation dynamics, water quality parameters, and microbial community composition. The results indicated no significant differences were observed between the ISs and AEPSs groups during the cultivation period. AEPSs significantly enhanced flocculation efficiency, achieving a final floc volume of 60 mL/L. This enhancement was attributed to the selective promotion of floc-forming microbial taxa, such as *Comamonas*, which can secrete procoagulants like EPS, and *Pseudomonas* and *Enterobacter*, which have denitrification capabilities. Water quality monitoring revealed that both AEPSs and AEPTs achieved nitrogen removal efficiencies exceeding 50% in the biofloc system, with AEPSs outperforming AEPTs. This is closely related to the fact that the microorganisms with increased flocculation contain numerous nitrifying and denitrifying bacteria. So, the intracellular polysaccharides were the key component of the ISs of *A. pyrenoidosa* that drive biofloc formation. These findings provide critical insights into the functional roles of algal-derived macromolecules in biofloc dynamics and their potential applications in wastewater treatment.

## 1. Introduction

Over the past few decades, bioflocculants have garnered significant attention in environmental engineering owing to their sustainable and eco-friendly properties. These substances, predominantly produced by microorganisms such as algae, yeast, and bacteria, derive their flocculation capabilities from biopolymers synthesized during cellular growth or localized on microbial cell surfaces [1]. Key functional components include polysaccharides [2], proteins [3], and nucleic acids [4], which collectively mediate particle aggregation through mechanisms such as charge neutralization, bridging effects, and hydrophobic interactions. Notably, Li et al. [2] demonstrated their efficacy in removing suspended contaminants from wastewater, including dye pigments, heavy metal ions, and other pollutants [5], highlighting their versatility in environmental remediation applications.

As a high-protein microalga, *Auxenochlorella pyrenoidosa* (formerly *Chlorella pyrenoidosa*) not only has a high self-flocculation rate [6] but also exhibits high nitrogen removal efficiency in wastewater [7]. Microalgae preferentially take up ammonia nitrogen, which is the main component of total nitrogen (about 60%) and the most important pollutant in wastewater, and convert it to substances for growth through nitrogen metabolism [8]. The study by Chen et al. [9] confirmed that the highest biofloc formation was observed in *A. pyrenoidosa* at algal concentrations of 5 × 10^9^ and 10^10^ cells/L. Polysaccharide-derived biopolymeric flocculants have emerged as sustainable alternatives for contamination mitigation, demonstrating exceptional effectiveness in removing suspended particulates, organic pollutants, and chromatic compounds across various aqueous systems. Their environmental compatibility, resulting from inherent biodegradation pathways, combined with multifunctional adsorption capabilities, has facilitated cross-sector adoption in industrial effluent management, mineral recovery processes, and bioproduct manufacturing sectors [10]. Algal extracellular polysaccharides (EPSs) have gained significance in promoting self-flocculation or as bioflocculants [11]. Consequently, these biopolymeric agents exhibit dual-phase applicability in phytogenic biomass valorization and industrial effluent remediation, maintaining eco-compatibility by minimizing the release of hazardous by-products.

Polysaccharide-based particles have been studied due to their safety, stability, biocompatibility, biodegradability, and hydrophilicity [12]. The addition of an appropriate amount of fucoidan into animal feeds can enhance intestinal health [13] and improve production performance [14]. Polysaccharide bio-based flocculants can be classified into three main categories based on their sources: (i) algal polysaccharide flocculants; (ii) microbial polysaccharide flocculants; and (iii) plant polysaccharide flocculants [15]. These natural flocculants are primarily derived from three biological sources: crustacean waste, agroforestry materials, and microbial organisms [16]. Cellulose and starch-based flocculants are typically sourced from agricultural and forestry by-products, while alginate, chitosan, and microbial polysaccharides are generally extracted from marine algae, crustacean exoskeletons, and various microorganisms [17]. Notably, microbial communities—including bacteria, fungi, and algae—represent a particularly important source of bioflocculant polysaccharides due to their metabolic diversity and sustainable production potential [18]. The cohesive interaction between suspended particulates and cellular entities mediated by polysaccharide-based flocculants arises from two synergistically operating physicochemical pathways: electrostatic charge neutralization through ionic interactions and macromolecular bridging via polymer chain elongation [19]. Charge neutralization occurs when oppositely charged polysaccharide flocculants adsorb onto colloidal particles. This adsorption significantly reduces the surface charge density through electrostatic interactions, thereby inducing particle destabilization by diminishing electrical double-layer repulsion [20,21]. Consequently, weakened electrostatic repulsion allows van der Waals forces and other attractive interactions to dominate, facilitating particle aggregation. Optimal functionality of this interfacial charge interaction is achieved with oligomeric polysaccharide derivatives, where strong electrostatic attraction between anionic polymer moieties and cationic surface domains achieves >90% colloidal destabilization efficiency under dynamic shear conditions.

Substantial experimental evidence has established that EPSs from microalgae play a pivotal role in enhancing biofloc formation. Notably, algal intracellular polymeric substances (IPSs) demonstrate remarkable compositional congruence with extracellular secretory products (ESPs), while exhibiting significantly higher concentrations than their extracellular counterparts [22]. This chemical similarity, coupled with the quantitative dominance of IPSs, strongly suggests functional parallels between intracellular and extracellular polysaccharides and proteins in algal systems. The observed compositional isomorphism implies potential mechanistic synergies in bioflocculation processes, where both IPSs and ESPs may engage in analogous molecular interactions through shared functional groups and charge characteristics. The compositional superiority of IPSs is fundamentally rooted in their structural complexity. Polysaccharide components within IPSs, including starch and glycoproteins, exhibit significantly higher molecular weights and a greater abundance of reactive functional moieties (e.g., hydroxyl and carboxyl groups) than their extracellular counterparts [23]. This enhanced macromolecular architecture facilitates superior flocculation performance through two synergistic mechanisms: augmented bridging capacity via extended polymer chains and intensified charge neutralization effects through multiple electrostatic interaction sites. In studying the efficiency of *Curvularia moringae* J-26 (flocculating fungi) in flocculating *Chlorella* sp., Jing et al. found that protein-like components in the EPSs contributed to the flocculation efficiency [24]. Similarly, when extracting bioflocculants produced by *Cobetia marina* L03 (microbial flocculant), Lei et al. detected the presence of proteins within these substances [25]. Similarly, proteins extracted from microalgae may possess analogous capabilities. When Aljuboori lyophilized the bioflocculant derived from *Scenedesmus quadricauda* (green algae) biomass into a powder form, compositional analysis revealed that carbohydrates and proteins constituted 56.7% and 41% of its mass fraction, respectively [26]. This evidence suggests that proteins likely contribute to flocculation promotion. Furthermore, the proteinaceous constituents of IPSs—particularly tryptophan-rich polypeptides—demonstrate exceptional flocculation-enhancing properties. These biomolecules interact with suspended particulates through a dual-action mechanism that combines hydrophobic interactions and electrostatic forces, thereby forming stable supramolecular architectures that significantly enhance floc formation kinetics and structural integrity [27]. The pronounced flocculation activity of IPS-associated proteins is further reinforced by their conformational flexibility, enabling adaptive binding to heterogeneous particle surfaces.

Our earlier study [28] demonstrated that the intracellular substances (ISs) of *A. pyrenoidosa* significantly influence biofloc formation, with polysaccharides and proteins being the two main components. Despite these insights, systematic investigations into the isolated effects of IPS-derived polysaccharides and proteins on biofloc formation are scarce. To identify the key components of the ISs of *A. pyrenoidosa* that drive biofloc formation, we assessed the flocculation contributions of these two main ISs components by adding IPS-derived polysaccharides and proteins. The results of this study provide data support for some of the gaps in bioflocculation technology and offer a new strategy for algal biomass recovery and sustainable wastewater management.

## 2. Materials and Methods

### 2.1. Biological Materials

*A. pyrenoidosa* (FACHB-9), sourced from the Institute of Hydrobiology, Chinese Academy of Sciences in Wuhan, was utilized in this study. *Bacillus subtilis* (*B. subtilis*), obtained from Beijing BioWorks Co., Ltd. (Beijing, China), was inoculated into an LB liquid medium, which was autoclaved at 121 °C for 20 min. Post-inoculation, the cultures were incubated for 24 h at 30 °C with a shaking speed of 150 revolutions per minute (rpm) in a shaking incubator.

### 2.2. Experimental Design for Extraction of ISs, AEPSs, and AEPTs

The ISs were isolated from *A. pyrenoidosa*. The polysaccharides and proteins, extracted from the ISs of *A. pyrenoidosa,* were defined as algal-extracted polysaccharides (AEPSs) and algal-extracted proteins (AEPTs), respectively. First, a sufficient quantity of algal solution was cultivated to achieve a cell density of 5 × 10^9^ cells/L, which has been reported to be optimal for biofloc formation [9]. The algal cells were then harvested by centrifugation at 5000 rpm for 15 min at 4 °C and subsequently washed with ultrapure water. After thorough washing, the harvested cells underwent three freeze–thaw cycles (−80 °C for freezing and 25 °C for thawing) followed by ultrasonic disruption. The disruption solution was then filtered through a 0.45 μm filter membrane to remove cellular debris, yielding the ISs solution.

Second, the AEPSs were extracted and purified using the Sevage method [29]. The specific extraction steps are as follows:Homogenized intracellular components of *A. pyrenoidosa* were centrifuged at 3000 rpm for 20 min at 4 °C. The supernatant was transferred via pipette into a dried beaker.The pellet was subjected to triplicate repetitions of Step a, with all supernatants pooled.The combined supernatant was stored overnight at −80 °C, concentrated to 40 mL by lyophilization, then mixed with 10 mL chloroform-n-butanol (4:1 *v*/*v*) under vigorous vortexing. After phase separation, the upper phase was collected and reapplied to a fresh chloroform-n-butanol solution. This partitioning procedure was repeated twice.Eighty milliliters of 95% (*v*/*v*) ethanol were added to the processed supernatant. Following 24 h static incubation at 4 °C, the upper phase was discarded. The lower phase was transferred to sterilized centrifuge tubes, stored overnight at −80 °C, and lyophilized. The resultant material was sequentially washed thrice with 10 mL aliquots of absolute ethanol, diethyl ether, and acetone, respectively. After oven-drying, purified polysaccharides were obtained and stored at −80 °C.

Third, the acetone precipitation method was used to extract AEPTs [30]. The specific extraction steps are as follows:Intracellular components of A. pyrenoidosa were centrifuged at 6500 rpm for 20 min at 4 °C. The supernatant was collected and stored at 4 °C for subsequent use.The pellet was resuspended in 10 mL of 10 mmol/L phosphate-buffered saline (PBS, pH 7.2). After combining supernatants from triplicate extractions, this procedure was repeated three times.Two volumes of acetone were added to the combined supernatant, followed by overnight incubation at −20 °C. The mixture was centrifuged at 6500 rpm for 20 min at 4 °C. The resultant pellet was stored at −80 °C overnight, lyophilized in a vacuum freeze-dryer, and the powder was preserved at −80 °C for further use.

### 2.3. Construction of Biofloc System and Design of Measurement Index

The polysaccharide content in the ISs solution was quantified using a phenol-sulfuric acid method with a spectrophotometer [31], while the protein content was determined using the Coomassie brilliant blue method [32]. Quantitative analysis revealed that the theoretical contents for polysaccharides and proteins in the ISs solution were 2.16 mg and 0.37 mg, respectively. However, subsequent application of the steps in Section 2.2 yielded purified AEPSs and AEPTs with purities of 35.7% and 37.0%, respectively. To achieve the actual measured contents, the practical extracted masses should correspond to the final isolated masses of 6.05 mg for AEPSs and 1.00 mg for AEPTs. Therefore, we designed three experimental groups: ISs, AEPSs, and AEPTs. Tap water (TW) was used as the control group (Table 1).

*B. subtilis*, recognized as an effective bioflocculant [33], significantly promotes biofloc formation. To accelerate the initiation speed of biofloc formation, 2 × 10^7^ CFU/L (Colony Forming Unit/Liter) of *B. subtilis* was initially added per container, as reported to be effective by He et al. [34]. The C/N was controlled at 15:1; the carbon source was glucose (150 mg/L); and the nitrogen source was urea (10 mg/L). Glucose was supplemented every 3 days to a C/N of 15:1, based on the TN measured every 3 days.

To ensure consistent environmental conditions, all the tanks were positioned in a greenhouse, where the water temperature (T) was strictly regulated at 26 ± 1 °C. A natural light–dark cycle of 12 h of light followed by 12 h of darkness was maintained for both the experimental and control groups. The pH levels were adjusted to remain within the range of 7.0 to 8.0 using sodium carbonate, while dissolved oxygen (DO) levels were sustained between 6.0 and 9.0 mg/L with the assistance of an air pump. These critical environmental parameters—T, pH, and DO—were monitored daily throughout the 13-day experimental period. No water was exchanged during this period; instead, any water loss due to sampling and evaporation was replenished.

Aerated water samples were initially analyzed to establish baseline values. Following cultivation, samples were collected at intervals on days 1, 4, 7, 10, and 13. Each sample was analyzed for biofloc volume (FV), total suspended solids (TSSs), turbidity (Turb), total ammonia nitrogen (NH_4_^+^-N), nitrite nitrogen (NO_2_^−^-N), nitrate nitrogen (NO_3_^−^-N), and total nitrogen (TN). On days 7 and 13, additional samples were collected for the analysis of microbial communities in the aquatic environment.

### 2.4. Determination Methods

To assess the settling volume of bioflocs, 1 L of the well-mixed water was collected in an Imhoff cone [35]. After allowing the sample to settle undisturbed for 30 min, the volume of the precipitated bioflocs was recorded as FV, expressed in mL/L [36]. For determining TSS, 50 mL of the sample was employed using the gravimetric method. Turbidity was assessed using a turbidimeter (wtw430) with 20 mL of the sample. T, pH, and DO were determined using the electrode method with a ProPlus YSI meter. TN, NH_4_^+^-N, NO_2_^−^-N, and NO_3_^−^-N were detected using an automatic flow injection analyzer (QC8500, HACH, Loveland, CO, USA). Specifically, TN was measured using the flow injection analysis method with naphthyl ethylenediamine dihydrochloride (HJ 668-2013). NH_4_^+^-N was determined utilizing the flow injection spectrophotometric method involving salicylic acid (HJ 666-2013). Concurrently, NO_2_^−^-N and NO_3_^−^-N were measured by flow injection analysis method, as described by Kazemzadeh and Ensafi [37]. During multiple pre-experiments, the glucose consumption of the bioflocculation system was determined every 3 days by a total organic carbon analyzer, almost reaching the level of complete consumption. In order to reduce the error caused by the length of time of the TOC determination, the amount of glucose supplemented every three days depended on the concentration of TN determined, so that the C/N in the system was maintained at the level of 15:1.

### 2.5. Microbial Community Analysis

A 100 mL aliquot of water was filtered through a 0.22 μm membrane and stored at −80 °C. DNA extraction, amplification, and sequencing were conducted by Majorbio Bio-Pharm Technology Co., Ltd. (Shanghai, China). The abundance of the 16S rDNA gene was analyzed by sequencing using Illumina sequencing technology. For bacterial analysis, forward primers 338F (5′-ACTCCTACGGGAGGCAGCAG-3′) and 806R (5′-GGACTACHVGGGTWTCTAAT-3′) amplified the V3-V4 hypervariable region of the 16S rRNA gene by a T100 Thermal Cycler PCR thermocycler (BIO-RAD, Hercules, CA, USA). In phylogenetic and population genetic studies, an Operational Taxonomic Unit (OTU) refers to a standard classification unit employed to simplify taxonomic labeling during analysis. The experimental results are presented at the phylum and genus levels, with OTUs clustered via a 97% similarity threshold.

### 2.6. Statistical Analysis

The analysis encompassed phylum and genus levels. Bacterial richness was estimated via the Chao1 index, while diversity was assessed via the Shannon and Simpson indices. Principal coordinates analysis (PCA) was applied to assess dissimilarity in species composition among the groups, followed by linear discriminant analysis (LDA) to identify the representative genera contributing to differences between the experimental and control groups. A higher LDA score signifies a greater impact of a species’ abundance on the observed differences [38].

Statistical analysis was conducted employing the IBM SPSS Statistics 26.0 software. Data analysis was executed using the univariate analysis of variance (ANOVA). A significance level of *p* < 0.05 was set to denote substantial disparities in outcomes, with results presented as mean ± standard deviation (mean ± SD).

For data derived from high-throughput sequencing techniques, the Kruskal–Wallis rank sum test was utilized for statistical analyses. A significance level of *p* < 0.05 marked noteworthy intergroup variations, while *p* < 0.01 indicated highly significant disparities among groups.

## 3. Results

### 3.1. Formation and Characteristics of Biofloc Aggregates

The temporal effects of each group on biofloc formation are illustrated in Figure 1. With the exception of the TW group, the other groups exhibited varying increases in TSSs throughout the cultivation period. No significant differences in trends or growth were observed between the ISs and AEPSs groups during the cultivation period (*p* > 0.05), with a rapid increase observed by day 7, followed by a moderation and continued growth post-day 10. TSS levels in the ISs group consistently surpassed those in the AEPSs group. In contrast, the growth of the AEPTs group plateaued after 10 days, with formation amounts significantly lower than those of the ISs and AEPSs groups (Figure 1a). No significant differences in FV were observed between AEPTs and TW groups throughout the experiment (*p* > 0.05), with values approaching zero. Concurrently, the FV for the ISs and AEPSs groups continuously increased, showing no significant divergence from day 1 to day 7. Notably, by day 10, the FV levels of the ISs group markedly exceeded those of the AEPSs group, maintaining this distinction until the end of the experience (Figure 1b). The turbidity of the AEPSs group increased the most rapidly by day 4, while the ISs and AEPTs groups also exhibited rapid increases without a significant difference (*p* > 0.05). By day 7, the turbidity of the ISs group reached the highest level among all the groups, while the AEPSs and AEPTs groups had attained a comparable level. All three groups initially experienced an increase in turbidity, which then decreased from days 10 to 13, consistently following the pattern ISs > AEPTs > AEPSs. The turbidity of the TW group gradually increased during the cultivation period but remained lower than that of the others. In contrast, the turbidity of the ISs, AEPSs, and AEPTs groups increased steadily from day 1 to day 10, followed by varying degrees of decline on day 13 (Figure 1c).

### 3.2. Alterations in Water Quality

The effects of each group on TN and tri-state nitrogen in the biofloc system over time are shown in Figure 2. Throughout the cultivation period, the TN levels in each group gradually declined. On day 7, the TN content in the AEPSs group was lower than that in the other groups, and the TN content of the remaining breeding stages was TW > AEPTs > AEPSs > ISs. After the experiment, the ISs group achieved the highest TN removal rate of 69%, followed by the AEPSs group at 65% and the AEPTs and TW groups at 50% each (Figure 2a). The content of NO_3_^−^-N in each group decreased rapidly in the first 4 days and then tended to be moderate. The content of the ISs and AEPSs groups was consistently lower than that of the AEPTs and TW groups, and there was no significant difference between them (*p* > 0.05) (Figure 2b). NO_2_^−^-N levels in the ISs group gradually increased from days 1 to 7 and declined to initial levels by day 7 to day 13. The AEPSs group showed stable NO_2_^−^-N levels from days 1 to 4 (*p* > 0.05), a peak at the highest levels observed across all groups by day 10, and a return to initial experimental levels by day 13. NO_2_^−^-N levels in the AEPTs group escalated from days 1 to 10 and experienced a notable decrease by day 13, yet they remained higher than those in other groups. The TW group showed a slight decline by day 4, followed by stabilization, with no significant changes by the end of the experiment (*p* > 0.05). By day 13, the NO_2_^−^-N levels were ordered as AEPTs > ISs = AEPSs > TW, all below 0.1 mg/L (Figure 2c). NH_4_^+^-N levels in the ISs group escalated from days 1 to 7 and then plateaued between days 7 to 13. The AEPSs group experienced a surge to the peak value of 1.9 mg/L among all groups by day 4, followed by a significant decrease by day 10, and a rebound to levels comparable to group ISs by day 13. Throughout the breeding period, the NH_4_^+^-N content of the AEPTs and TW groups gradually increased, with the AEPTs group consistently outperforming the TW group. At the end of the experiment, the content of NH_4_^+^-N in each group was ISs = AEPSs > AEPTs > TW group (Figure 2d).

### 3.3. Microbial Community Structure of Biofloc Aggregates

#### 3.3.1. Diversity Analysis

On day 7, the Ace and Chao1 values of the ISs group showed the highest population richness, while those for the TW group showed the lowest richness. By day 13, the ISs and AEPSs groups had higher microbial diversity according to the Shannon index, with the AEPTs group having the lowest diversity (Table 2).

On day 7, the number of the same OTUs in each group was 95, and the number increased to 144 on day 13, indicating an increasing trend toward a unified microbial community among the groups as culture time extended. In each group, the ISs group maintained the highest number of unique OTUs at the middle and end of the experiment. In contrast, the unique OTUs in the AEPSs and TW groups increased progressively over time, while those in the ISs and AEPTs groups gradually declined (Figure 3).

The results of PCA (Figure 4) showed that on day 7, the confidence intervals of each group did not overlap, and there were significant differences in microbial community structure among the groups (*p* < 0.05). On day 13, the confidence ellipses of the ISs, AEPSs, and AEPTs groups had a small area of overlap, and there was a certain difference between the two groups. At the same time, there was no overlap between the TW group and the other groups, and there was a significant difference (*p* < 0.05).

#### 3.3.2. Composition Analysis

At the phylum level, by day 7, the three dominant bacterial phyla across all groups were Proteobacteria, Bacteroidota, and Firmicutes, with Proteobacteria comprising over 80% (Figure 5a). By day 13, the top three dominant bacterial phyla in each group were Proteobacteria, Bacteroidota, and Myxococcota. The proportion of Proteobacteria in the ISs, AEPSs, and TW groups exceeded 80%, while the AEPTs group had less than 60%. Notably, the proportion of Bacteroidota in the TW group was over 35%, significantly higher than in the other groups (Figure 5b).

At the genus level, by day 7, the top five dominant bacterial genera across all groups were *Pseudomonas*, *Comamonas*, *Acinetobacter*, *Herbaspirillum*, and *Brevundimonas*. *Pseudomonas* was the most abundant in the ISs, AEPTs, and TW groups, while the second genus in the ISs group was *Flavobacterium*. The genus with the second highest proportion in the AEPTs group was *Brevundimonas*, and *Acinetobacter* was predominant in the TW group, constituting over 20%. *Herbaspirillum* was the most abundant genus in the AEPSs group, followed by *Comamonas* and *Pseudomonas* (Figure 5c). On day 13, the top five dominant bacteria in each group were *Pseudomona*, *Flavobacterium*, *Brevundimonas*, *Acinetobacter*, and *Comamonas*. In the ISs group, the proportion of *Pseudomonas* remained not significantly different from that observed on day 7 (*p* > 0.05). *Flavobacterium* proportions declined, while those of *Bacillus* and *Brevundimonas* increased. In the AEPSs group, proportions of *Pseudomonas* and *Brevundimonas* rose, while *Comamonas* and *Herbaspirillum* proportions fell. In the AEPTs group, *Flavobacterium* proportions surged by over 30%, *Pseudomonas* proportions diminished, and *Brevundimonas* proportions remained not significantly different from those on day 7 (*p* > 0.05). In the TW group, proportions of *Pseudomonas* and *Comamonas* fell, *Acinetobacter* exhibited the highest relative abundance, and *Brevundimonas* proportions increased (Figure 5d).

From day 7 to 13, the taxonomic changes at the phylum level for the ISs, AEPSs, and TW groups were minimal, while the relative abundance of the AEPTs group differed significantly. Microbial compositions in the ISs and AEPSs groups were rather stable at the genus level, but those in the AEPTs and TW groups showed notable shifts in both species and relative abundance.

*Acinetobacter*, *Candidatus*, and *Niabella* abundances in experimental groups ISs, AEPSs, and AEPTs significantly declined by day 7 in comparison with those in the control group (Group D), whereas *Aeromonas* and *Clostridium* abundances in the ISs group, *Herbaspirillum* and *Bacillus* abundances in the AEPSs group, and *Brevundimonas*, *Elstera*, *Leptothrix*, and *Vogesella* abundances in the AEPTs group increased significantly (*p* < 0.05) (Figure 6a).

By day 13, the abundances of *Acinetobacter*, *Sphingobium*, SM2D12, *Bacteriovorax*, *Haliangium*, and *Gemmobacter* in the experimental group significantly decreased, and the abundances of *Rickettsiaceae* in the ISs group were significantly reduced. *Herbaspirillum*, *Enterobacter*, *Microbacterium*, *Emiticicia*, *Rhodospirillales*, *Pajarorllobacter*, and *Solimonadaceae* in the AEPSs group and *Flavobacterium*, *Elstera*, and *Paracaedibacteraceae* in the AEPTs group significantly increased (*p* < 0.05) (Figure 6b).

The Spearman correlation analysis results (Figure 7) revealed significant correlations between *Pseudomonas* abundance and TN, NO_2_^−^-N, and FV on day 7 (*p* < 0.05). *Acinetobacter* showed significant correlations with TN, NO_2_^−^-N, FV, and Turb (*p* < 0.05). *Herbaspirillum*’s abundance was significantly linked to all the assessed water quality and floc formation indices (*p* < 0.05). The FV index saw significant correlations with both *Brevundimonas* and *Nubsella* (*p* < 0.05), while *Enterobacter*’s abundance was tied to TN and the TSS index (*p* < 0.05). *Aeromonas* exhibited significant correlations with a range of indices, including TN, NO_2_^−^-N, TSSs, FV, and Turb (*p* < 0.05). *Chryseobacterium*’s presence was significantly associated with NO_3_^−^-N and TN (*p* < 0.05), *Novosphingobium* with the FV index (*p* < 0.05), and the group *Allorhizobium-Neorhizobium-Pararhizobium-Rhizobium* with the Turb index (*p* < 0.05).

On day 13, *Acinetobacter* and *Herbaspirillum* were significantly correlated with all indices except NO_2_^−^-N (*p* < 0.05), and *Flavobacterium* showed significant correlations with TN and NH_4_^+^-N (*p* < 0.05). *Comamonas* was linked to NO_2_^−^-N (*p* < 0.05), while *Enterobacter* and *Microbacterium* were associated with NO_3_^−^-N, TN, NH_4_^+^-N, TSSs, and FV (*p* < 0.05). *Allorhizobium-Neorhizobium-Pararhizobium-Rhizobium* was correlated with NO_3_^−^-N and TSSs (*p* < 0.05); *Sphingobium* with TN, NH_4_^+^-N, TSSs, FV, and Turb (*p* < 0.05); *Legionella* with NO_3_^−^-N, TN, TSSs, and Turb (*p* < 0.05); *Xanthobacter* and *Asticcacaulis* with NO_3_^−^-N, TN, TSSs, and FV (*p* < 0.05); and *Enticicia* with TN and FV (*p* < 0.05).

## 4. Discussion

### 4.1. Water Quality

Extracellular polysaccharides (EPSs) are complex mixtures of polymers secreted by microbial cells, primarily composed of polysaccharides and protein-like substances [39]. In recent years, the biodegradation of EPSs has garnered increasing attention due to the EPSs’ role as a reservoir of carbon sources that enhances denitrification, particularly under nutrient-limited conditions [40]. Given that IPSs contains higher concentrations of polysaccharides and protein-like materials than EPSs, we hypothesized that IPSs may exhibit an even greater potential to promote denitrification processes.

Based on the observed changes in NO_3_^−^-N levels, the initial concentrations of NO_3_^−^-N in all experimental groups remained relatively high (approximately 10 mg/L), which is conducive to the growth of denitrifying bacteria [41]. A specific concentration of *B. subtilis*, renowned for its heterotrophic nitrification and aerobic denitrification properties, was introduced at the beginning of the experiment to accelerate the development of bioflocs [42]. This addition resulted in a gradual decrease in NO_3_^−^-N concentrations and a corresponding increase in NO_2_^−^-N levels. The analysis of microbial composition revealed the presence of various denitrifying bacterial genera at different concentrations across all the groups on days 7 and 13. For instance, *Comamonas*, known for its ability to adapt to environments with a low carbon-to-nitrogen (C/N) ratio while maintaining high denitrification activity, secretes significant amounts of extracellular polymeric substances that can serve as alternative carbon sources [40]. Similarly, *Pseudomonas* is capable of performing aerobic denitrification of both NO_3_^−^-N and NO_2_^−^-N [43,44]. Additionally, *Enterobacter* exhibits remarkable heterotrophic nitrification–aerobic denitrification capabilities, demonstrating high efficiency in organic nitrogen removal [45]. These denitrifying bacteria were present in all the experimental groups at specific biomass levels, which correlate with the observed rapid reduction in NO_2_^−^-N concentrations in this study. Notably, *Enterobacter* was found to be abundant in the AEPSs group on day 7, and its biomass remained at significant levels in both the ISs group and the AEPSs group by day 14. This observation corresponds to the lower NO_3_^−^-N concentrations in these two groups than in the others.

Under conditions of a high C/N ratio, N can undergo dissimilatory nitrate reduction to ammonium (DNRA), resulting in its conversion to NH_4_^+^-N [46]. Analysis of the microbial composition revealed the presence of *Herbaspirillum* at specific concentrations in both the ISs group and the AEPSs group on days 7 and 13. This genus is recognized for its biological nitrogen fixation capabilities [47], which involve converting ammonia released into the environment by other microorganisms through digestion and denitrification processes into NH_4_^+^-N, thereby fixing it in the aquatic environment. This finding aligns with the observed higher accumulation of NH_4_^+^-N in both the ISs group and the AEPSs group in this study. The diazotrophic bacteria identified in this study not only convert inert dinitrogen (N_2_) into bioavailable N, establishing a closed loop across the atmosphere–water–biomass continuum, but also mitigate N_2_O emissions while reducing N pollution.

### 4.2. Biofloc Formation

The composition of the intracellular polymers in microalgae is similar to that of extracellular secretions, with the concentration of the intracellular polymers being significantly higher than that of the extracellular secretions [20,41]. This indicates that the intracellular and extracellular polysaccharides in microalgae share similar components and functions. In their investigation of the structural and bioactive properties of intracellular and extracellular polysaccharides [48], Dong et al. likewise demonstrated that intracellular polysaccharides (IPSs) exhibit superior immunomodulatory activity [49]. This capability promotes probiotic colonization and proliferation, aligning with the observed higher abundance of beneficial bacterial genera in the AEPSs group in the present study. In the study by Meng et al., dynamic viscoelastic behaviors of intracellular polysaccharide solutions were likewise observed to be slightly higher than those of extracellular polysaccharide solutions at 25 °C [50], leading to the superior adsorption capacity of intracellular polysaccharides. This finding thus corroborates that the AEPSs group in the present study exhibited higher floc formation yield. Yang et al. [51] demonstrated that the extracellular polysaccharides of *Scenedesmus acuminatus* exhibit bioflocculation activity, facilitating the recovery of algal biomass. Similarly, Aljuboori et al. [26] found that the extracellular polysaccharides of *Scenedesmus quadricauda* also display bioflocculant activity in algal biomass recovery, achieving a flocculation efficiency of up to 86.7%. These studies highlight the potential of algal extracellular polysaccharides, which utilize microalgae-derived extracellular polymeric substances as coagulant aids, to enhance algal harvesting. Given the compositional similarity between the extracellular secretions and ISs of microalgae, we hypothesized that AEPSs may also possess similar bioflocculation activity. In this study, the extracted polysaccharides of *A. pyrenoidosa* contributed significantly to biofloc formation, with the flocculation amount being the highest among all groups except the intracellular polymer group and the formation quantity far exceeding that of the other groups.

The bridging mechanism is recognized as the primary flocculation mechanism for polysaccharide-based flocculants with similar charges or neutral properties. This mechanism is driven by interactions between the functional groups of polysaccharide molecules and suspended particles, which include van der Waals forces, electrostatic interactions, hydrogen bonding, and even chemical reactions [10]. High-molecular-weight (HMW) polysaccharide flocculants significantly enhance bridging efficiency. HMW flocculants generally demonstrate superior performance compared with low-molecular-weight (LMW) flocculants, primarily due to their enhanced adsorption capacity on particle surfaces, stronger bridging capability, and higher flocculation efficiency [52,53,54]. The extended polymer chains of HMW flocculants facilitate multi-point attachment and more effective inter-particle network formation, contributing to their improved performance.

Li et al. [5] identified polysaccharides as the primary active components in the bioflocculant produced by *Arthrobacter* sp. B4. These polysaccharides facilitate flocculation by adsorbing onto particle surfaces, thereby reducing the surface charge density and promoting particle aggregation. Additionally, both Li [5] and Zhang [55] demonstrated that the flocculation efficiency of these bioflocculants is maximized within a pH range of 5.0 to 8.0, which aligns with the pH conditions optimized in the present study. Over the past 20 years, the development of environmentally friendly polymers has been studied and reported, including extracellular secretions produced by bacteria [56], alginate-based flocs [57], and various polysaccharides, which are frequently utilized as bioflocculants. The polysaccharides commonly employed as bioflocculants include sodium alginate, chitosan, cellulose, starch, pullulan, xanthan gum, and pectin [58].

Microalgae are known to secrete substantial amounts of polysaccharides into their growth medium during proliferation [59]. Notably, benthic cyanobacterial species, such as *Phormidium* and *Anabaenopsis circularis,* have been identified as producers of extracellular polysaccharides with significant flocculating activity. This activity facilitates mutual flocculation between cyanobacterial cells and clay particles in aquatic environments [60]. This phenomenon is particularly relevant to algal bloom dynamics, as demonstrated by Chen’s [61] investigation of *Aphanothece halophytica*. During bloom events, this halotolerant cyanobacterium releases copious amounts of polysaccharides that interact synergistically with suspended clay particles, resulting in increased water viscosity and turbidity [62,63]. Field observations support this mechanism, revealing that dense aggregations of *A. halophytica* cells preferentially occur in highly turbid water columns, where polysaccharide–clay interactions are most pronounced. Maruyama [64] demonstrated that polysaccharides, when combined with protamine, exhibit flocculation efficiency even when added independently. Avnimelech et al. [65] proposed that cations reduce the electrostatic repulsion between negatively charged microbial cells and clay particles, allowing these particles to come close enough to form bonds. Consequently, microbial cells and clay particles aggregate and precipitate. These findings align with the results of our experiment, in which the AEPSs group exhibited high flocculation activity, resulting in the formation of a substantial amount of bioflocs.

*Comamonas* was found to be abundant in the AEPSs group on day 7 and 13. This genus can secrete EPSs into the water, providing a carbon source that contribution to flocculation in aquatic environments [40]. We speculated that the significant flocculation observed in the AEPSs group is related to the high abundance of bacteria.

## 5. Conclusions

This study examined the effects of intracellular polysaccharides and proteins, extracted from the intracellular components of *A. pyrenoidosa*, on the formation, water quality, and microbial community composition in biofloc systems. The results demonstrated that the intracellular polysaccharides effectively promoted biofloc formation. Flocculation efficiency analysis revealed that the AEPSs group enhanced microbial community diversity within bioflocs and selectively enriched floc-forming microorganisms, such as *Comamonas*, compared with the AEPTs group. These synergistic effects significantly improved bioflocculation efficiency driven by polysaccharides. The similarity between the ISs group and the AEPSs group indicates that intracellular polysaccharides are the primary substances driving biofloc formation. The promoting effects of these polysaccharides and proteins are achieved by stimulating the growth of denitrifying bacterial genera, such as *Pseudomonas* and *Enterobacter*. Consequently, these biopolymers achieved over 50% nitrogen removal efficiency in high-C/N-ratio water systems. Notably, the AEPSs group exhibited significantly higher nitrogen removal rates than the AEPTs group (*p* < 0.05). Integrating algal-extracted polysaccharides into aquaculture systems offers dual circular economy benefits: it not only reduces nitrogen loading to achieve water conservation but also stimulates probiotic proliferation, enabling the partial substitution of commercial carbon sources within such systems.

## Figures and Tables

**Figure 1 microorganisms-13-01704-f001:**
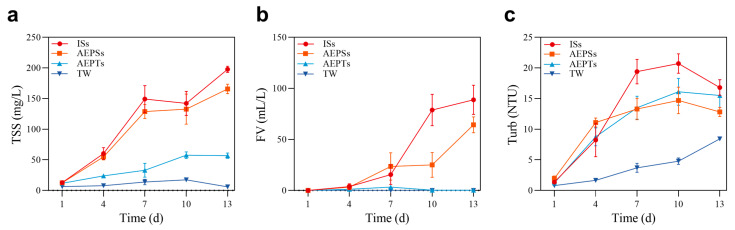
Effects of algal-extracted polysaccharides and algal-extracted proteins on biofloc formation ((**a**): TSSs; (**b**): FV; (**c**): Turb).

**Figure 2 microorganisms-13-01704-f002:**
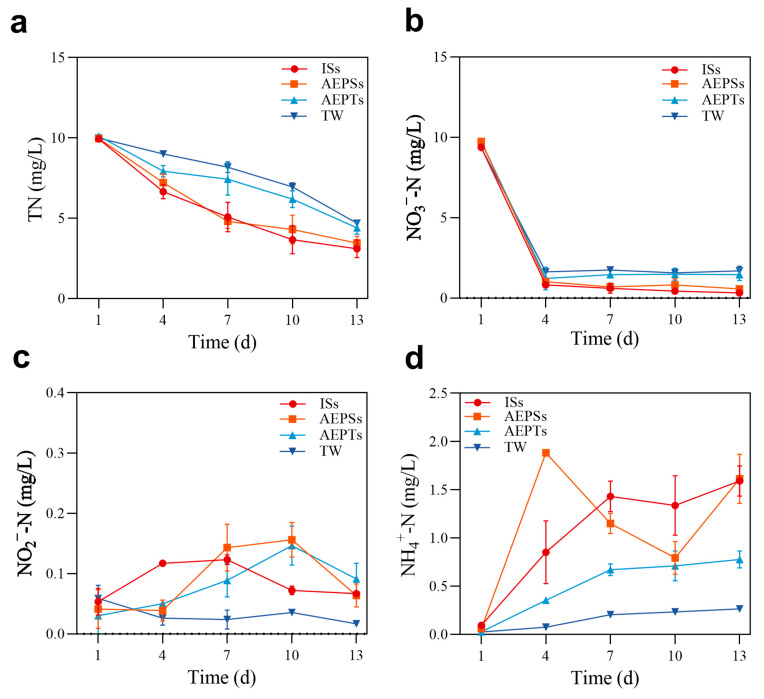
The temporal changes of TN and tri-state nitrogen during the biofloc system ((**a**): total nitrogen (TN); (**b**): nitrate nitrogen (NO_3_^−^-N); (**c**): nitrite nitrogen (NO_2_^−^-N); (**d**): ammonia nitrogen (NH_4_^+^-N)).

**Figure 3 microorganisms-13-01704-f003:**
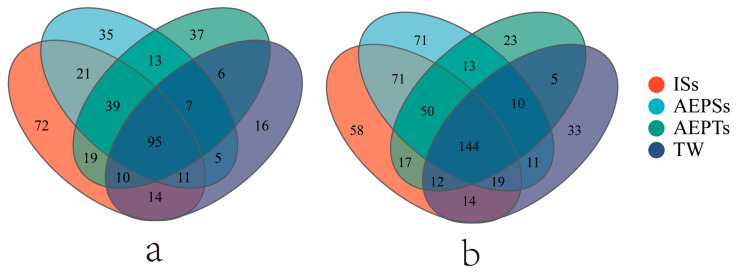
The Venn diagram of each group on day 7 (**a**) and day 13 (**b**) of the biofloc system.

**Figure 4 microorganisms-13-01704-f004:**
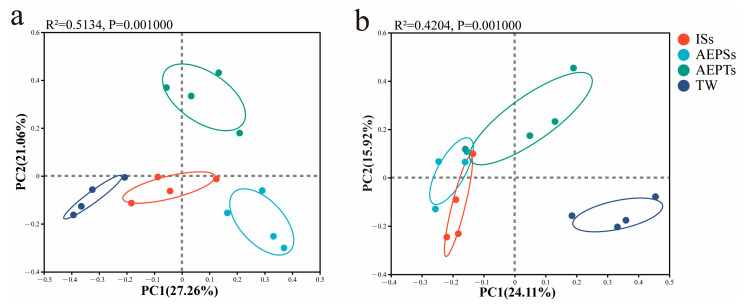
Principal component analysis (PCA) of microbial community function on day 7 (**a**) and day 13 (**b**) of the biofloc system.

**Figure 5 microorganisms-13-01704-f005:**
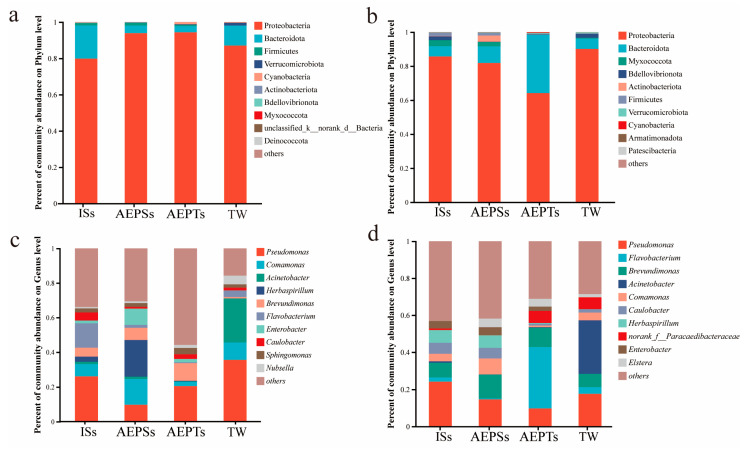
The relative abundance of microbial communities in each group on day 7 and day 13 ((**a**): day 7, phylum level; (**b**): day 13, phylum level; (**c**): day 7, genus level; (**d**): day 13, genus level).

**Figure 6 microorganisms-13-01704-f006:**
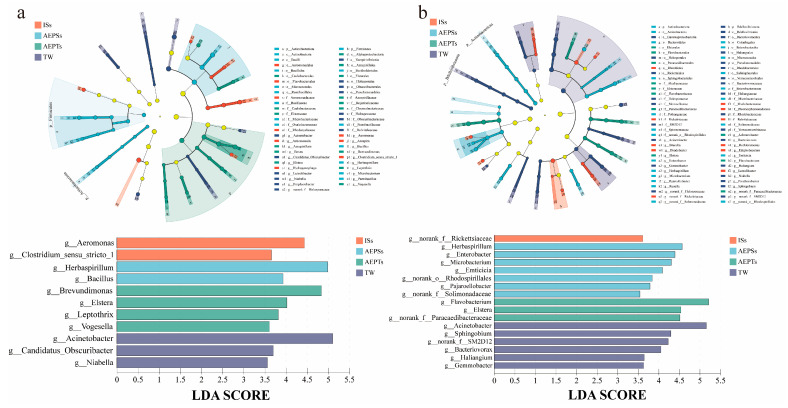
The biological community of the biofloc system on day 7 (**a**) and day 13 (**b**) based on the LEFSe analysis of the evolutionary branch map and the LDA value columnar distribution map. Note: A linear discriminant analysis (LDA) value > 3.5 of the flora, with statistically distinct biomarkers, is displayed in the histogram.

**Figure 7 microorganisms-13-01704-f007:**
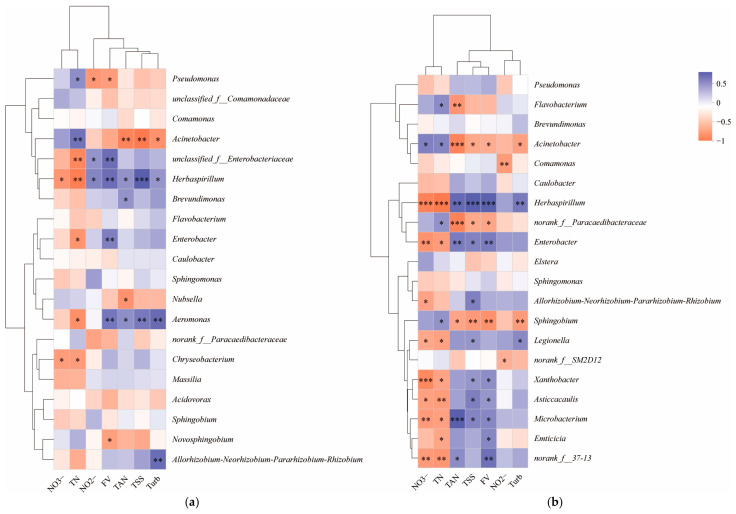
Spearman correlation analysis results of microbial genus level (relative abundance ranked top 20) and water quality and formation index on day 7 (**a**) and day 13 (**b**) of biofloc system formation. Note: * means *p* < 0.05, ** means *p* < 0.01, *** means *p* < 0.001; orange represents positive correlation, blue represents negative correlation, and the darker the color, the stronger the correlation.

**Table 1 microorganisms-13-01704-t001:** Experimental design of promoting the effects of polysaccharides and proteins in intracellular substances of *A. pyrenoidosa*.

Groups	ISs	AEPSs	AEPTs	TW (Control)
Microalgae intracellular components	Intracellular substances	Algal-extracted polysaccharides	Algal-extracted proteins	Tap water
Theoretical amount of extract added (mg)	2.16 + 0.37	2.16	0.37	/
Practical amount of extract added (mg)	/	6.05	1	/
*Bacillus subtilis* (CFU/L)	2 × 10^7^	2 × 10^7^	2 × 10^7^	2 × 10^7^
Nitrogen (mg/L)	10	10	10	10
C:N	15:1	15:1	15:1	15:1

Note: The value “2.16 + 0.37” for “Theoretical amount of extract added (mg)” in the ISs group corresponds to 2.16 mg of polysaccharides and 0.37 mg of proteins.

**Table 2 microorganisms-13-01704-t002:** Alpha diversity of the microbial community in each group on day 7 and day 13.

Time	Groups	Shannon	Simpson	Ace	Chao1	Coverage
Day 7	ISs	2.79 ± 0.36	0.12 ± 0.04	184.26 ± 21.61 ^a^	176.61 ± 27.95 ^a^	1.00
	AEPSs	2.26 ± 0.38	0.20 ± 0.10	147.61 ± 32.32 ^ab^	154.00 ± 34.19 ^ab^	1.00
	AEPTs	2.43 ± 0.40	0.16 ± 0.06	144.07 ± 24.52 ^ab^	137.88 ± 21.51 ^ab^	1.00
	TW	2.26 ± 0.21	0.19 ± 0.05	127.73 ± 5.66 ^b^	125.72 ± 6.43 ^b^	1.00
Day 13	ISs	3.03 ± 0.32 ^a^	0.11 ± 0.04	240.83 ± 40.51	236.97 ± 42.21	1.00
	AEPSs	2.97 ± 0.08 ^a^	0.10 ± 0.01	234.45 ± 37.39	231.18 ± 42.37	1.00
	AEPTs	2.19 ± 0.80 ^b^	0.27 ± 0.24	188.63 ± 36.94	183.95 ± 43.76	1.00
	TW	2.80 ± 0.22 ^ab^	0.13 ± 0.04	187.54 ± 3.49	192.06 ± 12.28	1.00

Note: The lowercase letters represent the significance of different groups at the same time.

## Data Availability

The original contributions presented in this study are included in the article. Further inquiries can be directed to the corresponding authors.

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
