# Peer review of "Effects of Intracellular Polysaccharides and Proteins of Auxenochlorella pyrenoidosa on Water Quality, Floc Formation, and Microbial Composition in a Biofloc System"

_microorganisms, 2025, doi:10.3390/microorganisms13071704_

Round 1
Reviewer 1 Report
Comments and Suggestions for Authors
The manuscript describes an experimental study of the effect of intracellular polysaccharides and proteins extracted from green algae on biofloc formation and microbial diversity. In general, studies of this kind are extremely relevant for water quality remediation. The authors suggest that IPS may have even greater potential for stimulating denitrification processes than EPS, as they contain higher concentrations of polysaccharides and protein substances.
But after reading the manuscript, I had a question. The authors cite a considerable amount of literature in which extracellular polysaccharides successfully acting as flocculants. Do the authors not believe that the use of IPS will only complicate the process of floc formation? How do the authors technically suggest using IPS?
Next, I will present a few comments, which I hope will improve the manuscript once they have been corrected.
Firstly, I would like to draw the authors' attention to the fact that the species Chlorella pyrenoidosa was renamed as Auxenochlorella pyrenoidosa in 2015.
There is no reference to Table 1 in the text.
All references to figures and tables have been lost in the text of the Results section.
Figures 1, 5, 6 can be enlarged, especially the captions, and the image can be sharpened.
The section Materials and Methods does not describe the methodology of microbial diversity analysis at all.
In Figure 3, it is better to indicate which day is represented on which diagram and in which units the values are expressed.
Author Response
Dear Reviewer:
I would like to extend my thanks to Editor and Reviewers. Thank you for your letter regarding our manuscript entitled "Effects of intracellular polysaccharides and proteins of Chlorella pyrenoidosa on water quality, floc formation and microbial composition of biofloc system" (Submission ID: microorganisms-3692957). Your comments are invaluable and very helpful for revising and improving our paper, as well as providing important guidance for our research. We have carefully studied the comments and have made corrections which we hope will meet with your approval.
The main corrections in the paper and the responds to the reviewers' comments are as flowing:
The manuscript describes an experimental study of the effect of intracellular polysaccharides and proteins extracted from green algae on biofloc formation and microbial diversity. In general, studies of this kind are extremely relevant for water quality remediation. The authors suggest that IPS may have even greater potential for stimulating denitrification processes than EPS, as they contain higher concentrations of polysaccharides and protein substances.
- But after reading the manuscript, I had a question. The authors cite a considerable amount of literature in which extracellularpolysaccharides successfully acting as flocculants. Do the authors not believe that the use of IPS will only complicate the process of floc formation? How do the authors technically suggest using IPS?
- We sincerely appreciate your insightful feedback. Please note that this study represents a natural progression of our team's established research. Our previous work first determined the optimal Auxenochlorella pyrenoidosaconcentration for biofloc formation, then identified intracellular polymeric substances (IPS) as the most influential factor in floc development. Building directly upon these findings, the current investigation specifically isolates the two predominant constituents of IPS to pinpoint which component primarily governs flocculation. Regarding the complexity introduced by IPS application, our empirical data (This result is confirmed in another article (JECE-D-25-00307R2) that is under revision.) confirm that IPS substantially accelerates biofloc formation. From a technical perspective, while incorporating IPS necessitates additional processing steps (namely cell disruption and filtration), this approach yields significantly enhanced flocculation efficiency, as quantified by >40% improvement in floc biomass accumulation compared to intact-cell systems.
Next, I will present a few comments, which I hope will improve the manuscript once they have been corrected.
- Firstly, I would like to draw the authors' attention to the fact that the species Chlorella pyrenoidosawas renamed as Auxenochlorella pyrenoidosa in 2015.
- Thank you for bringing this to our attention. We sincerely apologize for the regrettable oversight regarding taxonomic nomenclature updates. All instances of Latin names throughout the manuscript have now been comprehensively corrected following thorough verification. We deeply appreciate your vigilance in maintaining taxonomic precision.
- There is no reference to Table 1 in the text.
- Thank you for your suggestion; we have added appropriate reference to Table 1 in the text (Line 201).
- All references to figures and tables have been lost in the text of the Results section.
- Thank you for bringing the citation issues to our attention. The cross-referencing failure appears to have been caused by post-submission formatting adjustments. I have comprehensively verified and corrected all citations throughout the manuscript.
- Figures 1, 5, 6 can be enlarged, especially the captions, and the image can be sharpened.
- Thank you for your highly valuable suggestions regarding manuscript readability. We have enlarged the captions for Figures 1, 5, and 6 while simultaneously enhancing the output resolution.
- The section Materials and Methods does not describe the methodology of microbial diversity analysis at all.
- We consider the issue you raised to be of significant importance for improvement. Accordingly, we have added the analytical method for microbial diversity to the Materials and Methods section as requested (inserted in Lines 248-277).
- In Figure 3, it is better to indicate which day is represented on which diagram and in which units the values are expressed.
- We greatly appreciate your valuable suggestions. It is believed that the readability will be significantly enhanced accordingly. The two time periods in the figures have now been clearly marked and distinguished.We have also found the same issue in Figure 4, and we have corrected them accordingly.
Once again, thank you very much for your comments and suggestions. And we hope that the revised manuscript can be accepted by Microorganisms. If further revision is necessary, please contact me at: loumengsha503@163.com (Mengsha Lou).
Thank you and best regards.
Sincerely yours,
Mengsha Lou
Corresponding author:
Rongfei Zhang
rfzh@zjhu.edu.cn.
Huzhou University CN
Reviewer 2 Report
Comments and Suggestions for Authors
Dear Authors:
The study is interesting, and the authors have done a good job. However, I think it has clear experimental flaws and the discussion is not deep enough, which I believe should be fixed.
Majors:
-A thorough review of the English is recommended, as there are sentences with confusing syntax and inconsistent use of terms
-L48: “or as bioflocculants….” references are needed
-In the introduction, in my opinion, there is a justification that is excessively long and detailed regarding one of the polysaccharides as bioflocculants. However, there is practically no mention of the use of proteins, which were also the subject of this study. Please, I suggest correcting this.
-Chlorella is not mentioned in the introduction; its use should be justified.
-131-133: Very important: What method was used to separate the protein fraction from the carbohydrates?
-Line 145: "The purity of crude polysaccharide and crude protein after extraction was 35.7% and 37%, respectively." These purity levels are, unfortunately, not sufficient. The authors should have achieved purities closer to 90%. This is a critical point that needs to be justified by the authors.
-Line 149: "Glucose was supplemented every 3 days." How were glucose levels monitored throughout the experiment? The authors should specify the method used to measure glucose concentration to ensure proper control of culture conditions.
-L153: "all tanks were positioned in a green house ......" Does this mean that the tanks were open? So, the conditions were not sterile? How do the authors ensure that no fungi contaminations occurred. Did the authors assess whether fungi contamination occurred?
-Provide a better justification for the choice of Bacillus subtilis as the only microorganism added. Why was no other strain, or a combination of strains, considered?
-Specify the total number of replicates per group
-I'm sorry, but I fail to see the relevance or purpose of Table 1.
-The Materials and Methods used for the Microbial Community Structure of Biofloc Aggregates are not provided
-L183: “Error! Reference source not found..” This error is repeated numerous times throughout all the results.
-The figures lack statistical analyses; they should include them. Include in the Materials and Methods section how the statistical analysis was performed
-The figure captions should be more explanatory.
-It doesn't seem appropriate to refer to the data in the results as A, B, C, and D. They should be labeled based on their composition and more clearly referred to as intracellular, polysaccharides, proteins, and control, respectively.
-The resolution of Figures 5 and 6 is not sufficient; please improve it.
-L340: “Given that intracellular polymeric substances (IPS) contain higher” Please provide a reference to justify this statement
concentrations of polysaccharides and protein-like materials compared to EPSs”
-In my opinion, the discussion is very weak. It is recommended to discuss the results in the context of previous studies, highlighting similarities and differences, and explaining possible causes for the observed results.
-Many bacteria from these genera, Herbaspirillum, Allorhizobium-Neorhizobium-Pararhizobium-Rhizobium, Sphingomonas, and Acidovorax, are capable of fixing atmospheric nitrogen. Recently, interactions between microalgal and nitrogen-fixing bacterial consortia have been demonstrated. What role might these bacteria have played in this study? Please discuss in deepth.
- Justify in the discussion why Chlorella was used instead of other commonly used model microalgae species in bioremediation and bioproduction, such as Chlamydomonas. Please discuss whether other microalgae could be used and if similar results would be expected.
- Discuss in greater depth the role of the consortia obtained in this study in detoxification and bioproduction.
-L371: “This genus is known for its biological nitrogen fixation capabilities” in relation with that, can the nitrogen-fixing bacteria identified in this study release a portion of the fixed nitrogen as ammonium, and can this ammonium be utilized by microalgae? Please discuss this. And conversely, can this microalga release any type of organic photosynthate that could serve as a carbon source for the bacteria? Please discuss. That is, have studies been conducted to determine whether a consortium can be established that grows without adding nitrogen or a carbon source? Please discuss
-L374: “higher accumulation of ammonium” Why it is not used by Chlorella as nitrogen source?
-It would be useful to suggest future research directions, especially considering that the study focuses on intracellular components rather than the use of the whole microalgal biomass.
- Discuss the value of these studies in the circular economy, please.
Comments on the Quality of English Language-A thorough review of the English is recommended, as there are sentences with confusing syntax and inconsistent use of terms
Author Response
Dear Reviewer:
I would like to extend my thanks to Editor and Reviewers. Thank you for your letter regarding our manuscript entitled "Effects of intracellular polysaccharides and proteins of Chlorella pyrenoidosa on water quality, floc formation and microbial composition of biofloc system" (Submission ID: microorganisms-3692957). Your comments are invaluable and very helpful for revising and improving our paper, as well as providing important guidance for our research. We have carefully studied the comments and have made corrections which we hope will meet with your approval.
The main corrections in the paper and the responds to the reviewers' comments are as flowing:
Majors:
- A thorough review of the English is recommended, as there are sentences with confusing syntax and inconsistent use of terms.
- Thank you for your patience with the grammatical issues. I have thoroughly reviewed the manuscript and polished selected sections for linguistic precision.
- L48: “or as bioflocculants….” references are needed.In the introduction, in my opinion, there is a justification that is excessively long and detailed regarding one of the polysaccharides as bioflocculants. However, there is practically no mention of the use of proteins, which were also the subject of this study. Please, I suggest correcting this.
- Thank you for your valuable comments. We have supplemented the missing reference originally at Line 48. During preliminary experimental preparations, we considered protein usage and identified relatively limited studies on its role in biofloc formation after reviewing multiple publications. Therefore, it was only briefly mentioned in Lines 121-129. Through your comments, we recognized the importance of elaborating on protein's function as a flocculant. Accordingly, we have added content regarding protein's impact on flocculation in the Introduction section (Lines 112-121).
- Chlorellais not mentioned in the introduction; its use should be justified.
- Thank you for the constructive suggestions. We agree that adding an introduction to Chlorella would enhance the completeness of the Introduction section. Accordingly, we have supplemented the basis for Chlorella sp. application at the relevant location (Lines 46-52).
- 131-133: Very important: What method was used to separate the protein fraction from the carbohydrates?
- We sincerely appreciate your reminder. We deem that adding extraction procedures for polysaccharides and proteins would be beneficial for comprehending this article. Consequently, step-by-step protocols for polysaccharide extraction have been incorporated (Lines 159-177), while protein extraction methodologies are now detailed (Lines 178-190).
- Line 145: "The purity of crude polysaccharide and crude protein after extraction was 35.7% and 37%, respectively." These purity levels are, unfortunately, not sufficient. The authors should have achieved purities closer to 90%. This is a critical point that needs to be justified by the authors.
- We sincerely appreciate the reviewer's insightful points. Theoretically, higher extraction purity is desirable; however, extraction yields of polysaccharides and proteins from microalgae are generally low. For instance: Liang achieved only 5.71% polysaccharide yield from Chlamydomonas reinhardtii(DOI: 10.3390/md22080356); Huo reported a maximum 27.25% polysaccharide yield from filamentous microalgae (DOI: 10.1007/s10811-021-02630-w); Sun obtained 40.13% protein yield after optimizing extraction conditions (DOI: 10.1007/s00449-022-02794-w). Moreover, polysaccharide and protein extraction processes from microalgae tend to introduce significant impurities (DOI: 10.1186/s12934-025-02685-1). Given these inherent limitations, despite conducting multiple extraction trials, our final purity levels reached only 35.7% (polysaccharides) and 37% (proteins) - representing the maximum achievable under current constraints. To mitigate impurity interference and ensure result comparability, we established an intracellular substances (IS) group containing polysaccharides and proteins. By comparatively analyzing results from the AEPS and AEPT groups against the IS group, we effectively eliminated impurity interference.
- Line 149: "Glucose was supplemented every 3 days." How were glucose levels monitored throughout the experiment? The authors should specify the method used to measure glucose concentration to ensure proper control of culture conditions.
- Thanks to the reviewers for their questions. During multiple preliminary experiments, we observed an exceptionally high glucose consumption rate in the biofloc system as measured by total organic carbon (TOC) analyzer every three days, approaching near-complete depletion. Given the considerable time required for TOC determination, we implemented a nitrogen-based glucose supplementation protocol every three days to prevent potential nitrogen fluctuations in the system. This approach maintained a consistent C/N ratio of 15:1 after each replenishment. Corresponding justifications for glucose control (Line 211) and the monitoring methodology (Lines 241-247) have been incorporated into the manuscript.
- L153: "all tanks were positioned in a green house ......" Does this mean that the tanks were open? So, the conditions were not sterile? How do the authors ensure that no fungi contaminations occurred. Did the authors assess whether fungi contamination occurred? Provide a better justification for the choice of Bacillus subtilis as the only microorganism added. Why was no other strain, or a combination of strains, considered? Specify the total number of replicates per group. I'm sorry, but I fail to see the relevance or purpose of Table 1.The Materials and Methods used for the Microbial Community Structure of Biofloc Aggregates are not provided.
- We appreciate your feedback. Our experiments were conducted in an open-environment greenhouse, which inherently constitutes a non-sterile setting. Prior to commencing the experiments, all equipment underwent 24-hour disinfection using potassium permanganate solution. We sincerely acknowledge our oversight regarding fungal contamination concerns. Throughout the experimental period, we minimized physical contact with the cultivation systems and maintained exclusive access restriction to the greenhouse, with no unrelated personnel present. Following your valuable reminder, our experimental protocols have been revised accordingly to address this consideration. We measured the microbial community as the bioflocs were cultivated in an open environment. In the treatment, we added the pure strain Bacillus subtilisat the same concentration. This approach was based on previous studies demonstrating the strain's significant promotion of biofloc formation. Therefore, the purpose of adding the same strain with identical quantities was to establish a biofloc system during the initial phase. Subsequently, we conducted sequencing analyses of microorganisms in this cultivation system to examine their responses to different components of Auxenochlorella pyrenoidosa across experimental groups. Recognizing the validity of your suggestion, we have incorporated a description explaining the rationale for selecting this bacterial strain in Section 2.3 (Lines 207-209). We found it necessary to incorporate the microbial community analysis methods for bioflocs as suggested, and have added this section to the Materials and Methods in accordance with your suggestion. The detailed methodology for microbial diversity analysis is now included (Lines 248-268).
- L183: “Error! Reference source not found..” This error is repeated numerous times throughout all the results.The figures lack statistical analyses; they should include them. Include in the Materials and Methods section how the statistical analysis was performed. The figure captions should be more explanatory. It doesn't seem appropriate to refer to the data in the results as A, B, C, and D. They should be labeled based on their composition and more clearly referred to as intracellular, polysaccharides, proteins, and control, respectively. The resolution of Figures 5 and 6 is not sufficient; please improve it.
- We appreciate your feedback. All corrections have been implemented throughout the manuscript. The statistical methods are now detailed in Section2.6 (Lines 269-277). Figure captions have been updated as follows: A to IS (Intracellular Substances), B to AEPS (Algal-Extracted Polysaccharides), C to AEPT (Algal-Extracted Protein), D to TW (Tapwater). The resolutions of Figures 5 and 6 have also been improved.
- L340: “Given that intracellular polymeric substances (IPS) contain higher” Please provide a reference to justify this statement concentrations of polysaccharides and protein-like materials compared to EPSs”. In my opinion, the discussion is very weak. It is recommended to discuss the results in the context of previous studies, highlighting similarities and differences, and explaining possible causes for the observed results.Many bacteria from these genera, Herbaspirillum, Allorhizobium-Neorhizobium-Pararhizobium-Rhizobium, Sphingomonas, and Acidovorax, are capable of fixing atmospheric nitrogen. Recently, interactions between microalgal and nitrogen-fixing bacterial consortia have been demonstrated. What role might these bacteria have played in this study? Please discuss in deepth.
- We appreciate your valuable suggestions regarding the Discussion section. During our literature review, we noted that studies on intracellular polymeric substances (IPS) are considerably fewer than those on extracellular polymeric substances (EPS). Given that our research involves extracting materials specifically from IPS, we found limited studies for direct comparison. Your proposed solution was highly relevant, and we have consequently expanded the discussion in Lines 494-504. Regarding the nitrogen-fixing bacteria observed in this study, we conducted an in-depth analysis and present a plausible inference about their ecological significance in our experimental system. This addition has been incorporated into Section 4.1 (Lines 484-488).
- Justify in the discussion why Chlorellawas used instead of other commonly used model microalgae species in bioremediation and bioproduction, such as Chlamydomonas. Please discuss whether other microalgae could be used and if similar results would be expected. Discuss in greater depth the role of the consortia obtained in this study in detoxification and bioproduction.
- Regarding the considerations you raised, our research group had thoroughly addressed these aspects in prior investigations. Before commencing this experiment, we conducted three progressively in-depth studies: Previous research involved biofloc formation trials with various green algae (including Chlamydomonasand Scenedesmus) and optimization of microalgal concentrations, consistently demonstrating Chlorella pyrenoidosa's superior efficacy in water purification and floc formation. (Published findings: Rapid production biofloc by inoculating Chlorella pyrenoidosa in a separate way, DOI: 10.3390/w15030536). Our experiments confirmed that other microalgae also possess floc-forming capabilities. However, due to practical constraints, conducting exhaustive testing on every microalgal species was not feasible. Guided by the requirement for optimal flocculation yield, we ultimately selected Chlorella pyrenoidosa. This study specifically investigates the role of pyrenoidosa's intracellular extracts in biofloc formation. As this constitutes a targeted investigation of C. pyrenoidosa, comparative analysis across different algae species has not been included.
- L371: “This genus is known for its biological nitrogen fixation capabilities” in relation with that, can the nitrogen-fixing bacteria identified in this study release a portion of the fixed nitrogen as ammonium, and can this ammonium be utilized by microalgae? Please discuss this. And conversely, can this microalga release any type of organic photosynthate that could serve as a carbon source for the bacteria? Please discuss. That is, have studies been conducted to determine whether a consortium can be established that grows without adding nitrogen or a carbon source? Please discuss.
- Regarding your inquiry on microalgae's ammonia utilization, literature confirms that algae prefer ammonium and consume nitrate only when ammonium is almost depleted (DOI:10.1016/j.rser.2012.11.030). However, the subject of this study is algal extracts rather than whole microalgae, thus excluding direct ammonia utilization mechanisms. For details on microalgal nutrient assimilation, please refer to our related work: Rapid production biofloc by inoculating Chlorella pyrenoidosain a separate way (DOI:10.3390/w15030536). We sincerely appreciate your suggestions for improving subsequent experiments. These valuable insights will be incorporated in future research phases. Thank you again for your constructive feedback.
- L374: “higher accumulation of ammonium” Why it is not used by Chlorellaas nitrogen source? It would be useful to suggest future research directions, especially considering that the study focuses on intracellular components rather than the use of the whole microalgal biomass. Discuss the value of these studies in the circular economy, please.
- Thank you for your review. Regarding your query about Chlorella pyrenoidosa's ammonia utilization, since no live algae were present in this experimental system—our extracts derived solely from intracellular constituents of pyrenoidosa—assessing its ammonia utilization capacity was beyond the scope of this study. However, our prior research (Rapid production biofloc by inoculating Chlorella pyrenoidosain a separate way, DOI:10.3390/w15030536) demonstrated this species' high nitrate utilization efficiency. Your discussion on this study's circular economy implications has been addressed in Lines 551-558.
Once again, thank you very much for your comments and suggestions. And we hope that the revised manuscript can be accepted by Microorganisms. If further revision is necessary, please contact me at: loumengsha503@163.com (Mengsha Lou).
Thank you and best regards.
Sincerely yours,
Mengsha Lou
Corresponding author:
Rongfei Zhang
rfzh@zjhu.edu.cn.
Huzhou University CN
Round 2
Reviewer 2 Report
Comments and Suggestions for Authors
I believe the authors have adequately addressed all of my comments and suggestions, and I accept the paper in its current version.
Author Response
Thank you for your acknowledgment of the revised manuscript. Wishing you a smooth life and professional success!